# Spherical Subspace Potential Functional Theory

**Ágnes Nagy**

Department of Theoretical Physics, University of Debrecen, H-4002 Debrecen, Hungary; anagy@phys.unideb.hu

**Abstract:** The recently introduced version of the density functional theory that employs a set of spherically symmetric densities instead of the density has a 'set-representability problem'. It is not known if a density exists for a given set of the spherically symmetric densities. This problem can be eliminated if potentials are applied instead of densities as basic variables. Now, the spherical subspace potential functional theory is established.

**Keywords:** density functional theory; potential functional theory; subspace spherical densities

## 1. Introduction

Density functional theory (DFT) has developed into a very efficient tool to perform computations, especially in quantum chemistry, solid-state physics and materials science. Its success is due to the fact that this approach is based on the electron density instead of the many-electron wave function. The fundamental theorem of Hohenberg and Kohn [1] states that the ground-state electron density uniquely determines the external potential, and in principle it contains all information about the system. The Kohn–Sham version of DFT [2] provides powerful and accurate computation.

Recently, the density functional theory has been reshaped by Theophilou [3] using a set of the spherically symmetric densities instead of the density. This new version has been generalized [4,5] applying constrained search [6]. It turned out that there is a new representability problem, the 'set-representability problem' in this spherical theory, in addition to the usual v-representability problem. A set is representable if there is a density whose spherical averages around the nuclei yield the given set. Unfortunately, we are not aware of any easy way to resolve if a given set is representable. It is well-known that the v-representability problem of DFT can be eliminated using the potential as the basic variable instead of the density [7]. It has recently been discovered [8] that we can eliminate the set-representability problem if a set of the spherical potentials is taken instead of the set of the spherical densities as basic variable. This spherical potential functional theory (SPFT) has been extended to degenerate states [8].

The importance of using potentials has been known for a long time. The optimized potential method (OPM) [9,10] and several approximations [11–17] have turned out to be very useful in DFT calculations. The work of Yang, Ayers and Wu [7] (see also [18–20]) provided a firm foundation in the potential functional theory. These approaches can be extended to the spherical theory. SPFT gave a rigorous basis for them. Here, SPFT is utilized for subspaces and the spherical subspace potential functional theory (SSPFT) is established.

This paper is arranged as follows: The spherical subspace potential functional theory (SSPFT) is summarized in Section 2. The spherical subspace potentials are analyzed in Section 3. Section 4 is dedicated to the discussion. In the Appendix A, the proofs of the two main theorems of the theory are presented in case of the Coulomb external potential.

## 2. Spherical Subspace Potential Functional Theory

The spherical density functional theory and the spherical potential functional theory can be applied for non-degenerate as well as degenerate ground-states. In the degenerate case, it is better to use the subspace technique in both theories. This is due to the fact that

the subspace procedure has the advantage that the subspace density has the symmetry of the external potential if all degenerate eigenfunctions are taken into account with the same weight in constructing the subspace. Therefore, in case of degeneracy it is worth using the density matrix and the subspace density with the definitions

$$\hat{D} = \sum_{\gamma=1}^{g} \eta^{\gamma} |\Psi^{\gamma}\rangle \langle \Psi^{\gamma}| \tag{1}$$

and

$$\varrho = \sum_{\gamma=1}^{g} \eta^{\gamma} \varrho^{\gamma}, \tag{2}$$

where

$$\varrho^{\gamma} = N \int |\Psi^{\gamma}|^2 ds_1 d\mathbf{x}_2 \dots d\mathbf{x}_N \tag{3}$$

are the eigendensities corresponding to the wave functions $\Psi^{\gamma}$, and $g$ is the degree of degeneracy. The weighting factors $\eta^{\gamma}$ satisfy the conditions

$$1 = \sum_{\gamma=1}^{g} \eta^{\gamma} \tag{4}$$

and

$$\eta^{\gamma} \geq 0 . \tag{5}$$

This means that $\varrho$ can be constructed in many ways. In principle, any choice of $\eta^{\gamma}$ fulfilling Equations (4) and (5) can be used. However, there is a very special case, the one with equal factors $\eta^{\gamma}$. This situation provides subspace density having the symmetry of the external potential.

The spherical average of the subspace density $\varrho(\mathbf{r})$ (Equation (2)) with respect to the nucleus $\beta$ takes the form

$$\bar{\varrho}_\beta(r_\beta) = \frac{1}{4\pi} \int_{\Omega_\beta} \varrho(\mathbf{r}) d\Omega_\beta = \sum_{\gamma=1}^{g} \eta^{\gamma} \frac{1}{4\pi} \int_{\Omega_\beta} \varrho_\beta^{\gamma}(\mathbf{r}) d\Omega_\beta = \sum_{\gamma=1}^{g} \eta^{\gamma} \bar{\varrho}_\beta^{\gamma} , \tag{6}$$

where $r_\beta = |\mathbf{r} - \mathbf{R}_\beta|$ and $\Omega_\beta$ stands for the angles. $\mathbf{R}_\beta$ are the position vectors of the nuclei. $\bar{\varrho}_\beta^{\gamma}(r_\beta)$ is the spherical average of $\varrho^{\gamma}(\mathbf{r})$ with respect to the nucleus $\beta$ .

In case of a Coulomb external potential

$$v(\mathbf{r}) = - \sum_{\beta=1}^{M} \frac{Z_\beta}{r_\beta}, \tag{7}$$

where $M$ and $Z_\beta$ are the number and the atomic numbers of the nuclei, it can be proved that the set of the spherically symmetric subspace densities $\{\bar{\varrho}\}$, (e.g., $\bar{\varrho}_\beta(r_\beta)$, $\beta = 1, \dots, M$) uniquely determines the external potential if the external potential has the form of Equation (7) [3,5]. The proof is summarized in Appendix A.

This assertion is true for an even more general external potential

$$v(\mathbf{r}) = \sum_{\beta=1}^{M} v_\beta(r_\beta), \tag{8}$$

where each term $v_\beta$ in the sum depends only on the distance from the nucleus $\beta$ [4,5]. This theorem can be proved by the constrained search defining the functional

$$Q[\{\bar{\varrho}\}] = \min_{D \to \{\bar{\varrho}\}} tr\{\hat{D}(\hat{T} + \hat{V}_{ee})\}, \qquad (9)$$

where $\hat{T}$ and $\hat{V}_{ee}$ stand for the kinetic energy and the electron–electron energy operators.

The Euler equations

$$v_\beta(r_\beta) = -\frac{\delta Q}{\delta \bar{\varrho}_\beta}; \quad \beta = 1, \ldots, M \qquad (10)$$

can be obtained up to a constant, if $Q$ is functionally differentiable.

The potential functional approach is dual to the density functional formulation and yields a solution of the v-representability problem of the original DFT [7]. In the spherical density functional theory we have a set-representability problem in addition to the usual v-representability problem. It has recently been noted that this set-representability problem can also be avoided in the spherical potential functional theory [8].

In the subspace spherical potential functional theory (SSPFT), the set of the spherical potentials is the basic variable, not the set of the spherical densities. To stress it, the notation $\mathcal{E}$ is used to denote the energy functional instead of $E$ utilized in the subspace spherical density functional theory. Apparently,

$$\mathcal{E}_{\{v\}}[\{w\}] = tr\left\{\hat{H}_{\{v\}}\hat{D}_{\{w\}}\right\} = E_{\{v\}}[\{\bar{\varrho}\}_{\{w\}}], \qquad (11)$$

where

$$\hat{H}_{\{v\}} = \hat{H}_v = \hat{T} + \hat{V}_{ee} + \sum_{i=1}^{N} v(\mathbf{r}_i) \qquad (12)$$

is the Hamiltonian with the set of the external potential $\{v\}$. $\hat{D}_{\{w\}}$ is the ground-state density matrix in the external potential $\{w\}$ with the form of Equation (8). $\{w\}$ denotes the set $w_1, w_2, \ldots, w_M$. Obviously, the functionals $E_{\{v\}}$ and $\mathcal{E}_{\{v\}}$ should take the same value at the true ground-state.

The great advantage of the SSPFT is that there is no 'set-representability problem'; that is, there exists a potential for any set of the spherically symmetric potentials and for any potential there is a set of the spherically symmetric potentials provided that the potential is the form of Equation (8). The proof of this assertion is very simple for the Coulomb external potential and can be found in Appendix A. The more general case of Equation (8) is detailed in Ref. [8].

According to the variational principle, the ground-state energy is given by the minimum

$$E_{\{v\}} = \min_{\{w\}} \mathcal{E}_{\{v\}}[\{w\}] \qquad (13)$$

at the sole stationary point $\{w\} = \{v\} + \{c\}$, where $\{c\}$ stands for a set of arbitrary constants (see proof in [8].) The functional derivatives of $E_{\{w\}}$ yield the subspace spherical densities

$$\frac{\delta E_{\{w\}}}{\delta w_\beta(r_\beta)} = \bar{\varrho}_\beta(r_\beta). \qquad (14)$$

It is worth creating a non-interacting Kohn–Sham (KS) system in which computation can be realized. The non-interacting kinetic energy is given by

$$K[\{\bar{\varrho}\}] = \min_{\hat{D}_0 \to \{\bar{\varrho}\}} tr\{\hat{D}_0\hat{T}\}. \qquad (15)$$

The search is for all non-interacting density matrices $\hat{D}_0$ which yield the given set $\{\bar{\varrho}\}$. The Kohn–Sham equations have the form

$$\left[ -\frac{1}{2}\nabla^2 + v_s(\mathbf{r}) \right] \phi_i = \varepsilon_i \phi_i, \tag{16}$$

where the subspace density is

$$\varrho = \sum_{i=1}^{m} \lambda_i |\phi_i|^2. \tag{17}$$

The occupation numbers $\lambda_i$ can be fractional and the sum applies for all orbitals with a non-zero occupation number.

The Kohn–Sham potential is given by

$$v_s(\mathbf{r}) = \sum_{\beta=1}^{M} v_{s,\beta}(r_\beta), \tag{18}$$

where

$$v_{s,\beta}(r_\beta) = v_\beta(r_\beta) + v_{Hxc,\beta}(r_\beta); \quad \beta = 1, \ldots, M. \tag{19}$$

The Hartree plus exchange-correlation potential terms $v_{Hxc,\beta}(r_\beta)$ are defined as

$$v_{Hxc,\beta}(r_\beta) = \frac{\delta E_{Hxc}}{\delta \bar{\varrho}_\beta}; \quad \beta = 1, \ldots, M. \tag{20}$$

The Hartree and exchange-correlation functional $E_{Hxc}[\{\bar{\varrho}\}]$ is defined as

$$E_{Hxc}[\{\bar{\varrho}\}] = Q[\{\bar{\varrho}\}] - K[\{\bar{\varrho}\}]. \tag{21}$$

In some special cases it may be advantageous to utilize the partition

$$E_{Hxc}[\{\bar{\varrho}\}] = H[\{\bar{\varrho}\}] + E_x[\{\bar{\varrho}\}] + E_c[\{\bar{\varrho}\}], \tag{22}$$

that is, employing the sum of the Hartree (or classical Coulomb), the exchange and the correlation terms, respectively. The functional derivative provides the potential

$$v_{Hxc} = v_H + v_x + v_c \tag{23}$$

as a sum of the Hartree (or classical Coulomb), the exchange and the correlation potentials.

In the KS version of SSPFT, the true interacting energy is taken as a functional of the non-interacting potential

$$\tilde{\mathcal{E}}_{\{v\}}[\{w_s\}] = E_{\{v\}}[\{\bar{\varrho}\}_{\{w_s\}}], \tag{24}$$

where the tilde on $\mathcal{E}$ shows that this functional is different from $\mathcal{E}_{\{v\}}[\{w\}]$. (Of course, they should take the same value at the true ground-state).

The variational principle yields the true ground-state energy

$$E_{\{v_s\}} = \inf_{\{w_s\}} \tilde{\mathcal{E}}_{\{v\}}[\{w_s\}]. \tag{25}$$

Its stationary point corresponds to the solution of the KS equations.

### 3. Spherical Subspace Potentials

In the potential functional theory (PFT), the KS potential minimizes the total energy functional at the true ground-state. This procedure was known well before Yang, Ayers and Wu [7] developed the PFT and gave a firm foundation to it. The minimizing potential is generally referred to as the optimized effective potential (OEP). It was initiated by Sharp and Horton [9] using the Hartree–Fock method. For further generalizations see [10,21–25]. There exist several good approximations to OEP that proved to be more convenient for computation. The localized Hartree–Fock method (LHF) [11–13] has the favorable properties that it is invariant with respect to the unitary transformations of the orbitals, it is free of the self-interaction, and consequently, the potential has the correct long-range behavior. Another approximation to the OEP is the KLI (Krieger, Li and Iafrate) method [14–16]. The KLI approach is much simpler than the OEP and is more stable if a finite-basis-set is applied but it shows no invariance with respect to the unitary transformations of the orbitals.

Here, in SSPFT the total energy functional $\tilde{\mathcal{E}}_{\{v\}}[\{w_s\}]$ is minimized

$$\frac{\delta \tilde{\mathcal{E}}_{\{v\}}[\{w_s\}]}{\delta w_s} = 0 \tag{26}$$

at the true ground-state and the minimizing set $\{w_s\}$ provides the minimizing KS potential (18). This potential can be obtained using the OEP method as it can be used in the PFT theory. Approximations to the OEP approach, such as the LHF or the KLI procedures can also be applied. These methods can be employed if the energy is known as a functional of the orbitals. All these approaches can also include correlation. Now, the KLI technique is extended to the subspace potential functional theory. It is formalized so that it can contain correlation as well. An alternative derivation of the KLI potential presented earlier [17] is now refined. The idea is very simple. It is now summarized as follows. The KS equations can be written as

$$\left[-\frac{1}{2}\nabla^2 + w + w_H + w_{xc}\right]\phi_i = \varepsilon_i\phi_i, \tag{27}$$

where the Hartree plus exchange-correlation part of $w_s$ is the sum of the Hartree $w_H$ (the classical Coulomb) potential and the exchange-correlation $w_{xc}$ terms: $w_{Hxc} = w_H + w_{xc}$. As the energy is known or approximated as a functional of the orbitals, the functional derivative of the energy with respect to the orbitals leads to Hartree–Fock-like equations:

$$\left[-\frac{1}{2}\nabla^2 + w + w_H + \tilde{w}_{xc}^i\right]\psi_i = \epsilon_i\psi_i, \tag{28}$$

where

$$\tilde{w}_{xc}^i = \frac{\delta E_{xc}}{\psi_i \delta \psi_i^*} \tag{29}$$

is an orbital-dependent exchange-correlation potential. It is an operator; it is different for the different orbitals just like the Hartree–Fock exchange potential. If no correlation is taken into account, $\tilde{w}_{xc}^i$ reduces to an orbital-dependent exchange potential $\tilde{w}_x^i$ similar to the Hartree–Fock exchange potential.

After multiplying Equations (27) and (28) with $\phi_i^*$ and $\psi_i^*$, the sum for the occupied orbitals provides

$$\sum_{i=1}^{m}\lambda_i\phi_i^*\left[-\frac{1}{2}\nabla^2 + w + w_H + w_{xc}\right]\phi_i = \sum_{i=1}^{m}\lambda_i\varepsilon_i|\phi_i|^2 \tag{30}$$

and

$$\sum_{i=1}^{m} \lambda_i \psi_i^* \left[ -\frac{1}{2}\nabla^2 + w + w_H + \hat{w}_{xc}^i \right] \psi_i = \sum_{i=1}^{m} \lambda_i \epsilon_i |\psi_i|^2, \tag{31}$$

respectively. Using the approximation $\psi_i \approx \phi_i$ the difference of Equations (30) and (31) yields the exchange-correlation potential

$$w_{xc} = w_{xc}^S + v_{xc}^r, \tag{32}$$

where

$$w_{xc}^S = \frac{1}{\varrho} \sum_{i=1}^{m} \lambda_i \phi_i^* \hat{w}_{xc}^i \phi_i \tag{33}$$

is the Slater potential, and the remaining term is

$$w_{xc}^r = \frac{1}{\varrho} \sum_{i=1}^{m} \lambda_i (\varepsilon_i - \epsilon_i) |\phi_i|^2. \tag{34}$$

Equation (32) provides a KLI-like exchange-correlation potential and gives back the original KLI exchange potential if correlation is neglected. The detailed derivation can be found in Ref. [17].

While the subspace $\Sigma_0$ spanned by the non-interacting eigenfunctions $\Phi^\gamma$ ($\gamma = 1, \dots, g_0$) is unique, the non-interacting density matrix

$$\hat{D}_0 = \sum_{\gamma=1}^{g_0} \eta_0^\gamma |\Phi^\gamma\rangle\langle\Phi^\gamma| \tag{35}$$

depends on the weighting factors $\eta_0^\gamma$ satisfying relations

$$1 = \sum_{\gamma=1}^{g_0} \eta_0^\gamma \tag{36}$$

and

$$\eta_0^\gamma \geq 0 . \tag{37}$$

$g_0$ can be different from $g$, that is, the degree of the degeneracy in the non-interacting and the interacting systems might be different. However, of course, the densities and the sets of the spherically symmetric densities are the same in the real and the KS systems.

Any set of the weighting factors satisfying Equations (4), (5), (36) and (37) can be chosen. Certainly, the densities and the sets of the spherically symmetric densities depend on the selection of these factors. In principle, any of them is appropriate to build the theory. However, there exists an exceptional choice of the set. If all factors $\eta_0^\gamma$ are equal, the subspace density has the symmetry of the external potential [3]. This case is especially convenient for computation. To illustrate it, consider an atom with degenerate ground-state. Then the eigenfunctions are in most cases not spherically symmetric. Therefore, $\varrho^\gamma$ are not spherically symmetric either. Nevertheless, in several calculations the density is considered approximately spherically symmetric. If we use subspace density with equal weighting factors, this subspace density is exactly spherically symmetric. That is, the spherically symmetric approach is not an approximation, it is exact. Moreover, radial Kohn–Sham equations should be solved to produce this subspace density. This obviously means an immense gain in computation. For atoms, an earlier approach to treat multiplets with subspaces [26,27] is refined.

The radial subspace Kohn–Sham equations can be written as [5]

$$-\frac{1}{2}P_j'' + \frac{l_j(l_j+1)}{2r^2}P_j + w_s P_j = \varepsilon_j P_j \,,\tag{38}$$

where $P_j = rR_j(r)$ are the radial wave functions and $''$ stands for the second derivative. The subspace density takes the form

$$\sigma(r) = 4\pi r^2 \bar{\varrho}(r) = \sum_{j=1}^{m} \lambda_j (P_j)^2,\tag{39}$$

where $\lambda_j$ are the occupation numbers corresponding to the given configuration (see details of the derivation in Refs. [26,27]).

The exchange energy can be written as a functional of the orbitals. In a non-degenerate case it is the well-known Hartree–Fock expression as the non-interacting wave function is a single determinant. On the other hand, in cases of degeneracy, the KS wave functions are not single determinants. Still, the exchange energy is a functional of the $P_i$ (see e.g., [27,28])

$$E_x = E_x[P_i] = E_x^{av} + \sum_{k} C_k B_k[P_i] \,.\tag{40}$$

$E_x^{av}$ is the average exchange energy of the different multiplets associated with the configuration considered.

The B, C, N, O and F atoms are taken for illustration. In the cases of the B and F atoms, the electron configurations are $1s^2 2s^2 2p$ and $1s^2 2s^2 2p^5$, respectively. Both have the $^2P$ ground-state. The average exchange energy is taken as follows: $E_x = E_x^{av}$. Though the exact wave function is not spherically symmetric, the subspace density has this symmetry.

For the other atoms considered, there is only one term in the sum in Equation (40): $B_1[P_i] = F^2(pp)$. The exchange energies for the $1s^2 2s^2 2p^2$ (in the C atom) and the $1s^2 2s^2 2p^4$ (in the O atom) in electron configurations are

$$E_x(^3P) = E_x^{av} - \frac{3}{25}F^2(pp) \,,\tag{41}$$

$$E_x(^1D) = E_x^{av} + \frac{3}{25}F^2(pp) \,,\tag{42}$$

$$E_x(^1S) = E_x^{av} + \frac{12}{25}F^2(pp) \,.\tag{43}$$

That is, $C_1$ is equal to $-3/25$, $3/25$ and $12/25$ for $^3P$, $^1D$ and $^1S$, respectively. The exchange energies for the $1s^2 2s^2 2p^3$ electron configuration (N atom) are

$$E_x(^4S) = E_x^{av} - \frac{9}{25}F^2(pp) \,,\tag{44}$$

$$E_x(^2D) = E_x^{av} \,,\tag{45}$$

and

$$E_x(^2P) = E_x^{av} + \frac{6}{25}F^2(pp) \,,\tag{46}$$

i.e., $C_1$ is equal to $-9/25$, $0$ and $6/25$ for $^4S$, $^2D$ and $^2P$, respectively. $F^2(pp)$ is the Slater integral

$$F^2(pp) = \int \int R_{2p}^2(r_1)R_{2p}^2(r_2)\frac{r_<^2}{r_>^3}dr_1 dr_2 \,,\tag{47}$$

where $R_{2p}$ is the radial wave function of the 2p electrons (P = rR). $r_<$ stands for $r_1$ if it is smaller than $r_2$ and $r_2$ if it is smaller than $r_1$.

The subspace KLI method with Equation (40) for exchange and the local Wigner approximation [29]

$$E_c^{LW}[n] = \int \frac{a\bar{\varrho}}{b + r_s} d\mathbf{r} \,, \tag{48}$$

for correlation with the parameters $a = -0.02728$ and $b = 0.21882$ [30] are applied. $r_s$ is the Wigner–Seitz radius:

$$r_s = \left( \frac{3}{4\pi\bar{\varrho}} \right)^{1/3}. \tag{49}$$

Figures 1 and 2 present the subspace Hartree plus exchange-correlation potentials $w_{Hxc}$ for the N atom. Observe that the radial subspace KS equations (38) should be self-consistently solved for each multiplet. The subspace density $\sigma$ and radial wave functions $P_j$ are different for $^4S$, $^2D$ and $^2P$. The difference, however, is small. As the curves are very close, the part of Figure 1 where the differences are the biggest is enlarged and shown in Figure 2.

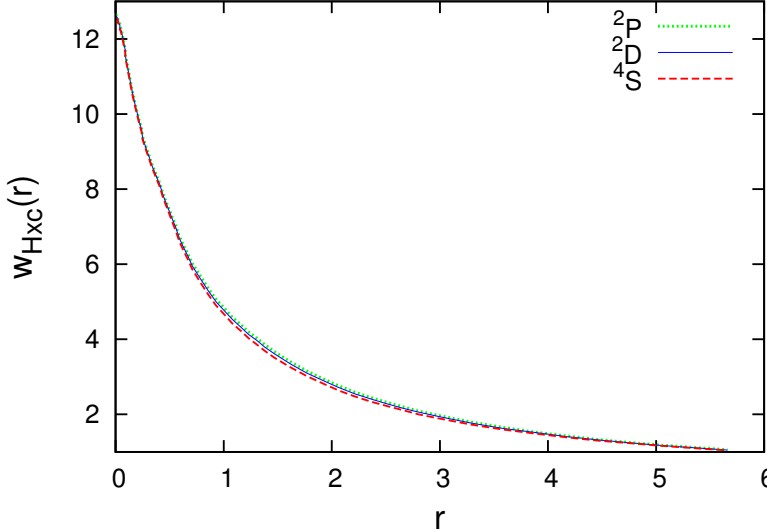

**Figure 1.** Subspace Hartree plus exchange-correlation potential for $^4S$, $^2D$ and $^2P$ of the N atom in atomic units (colored lines).

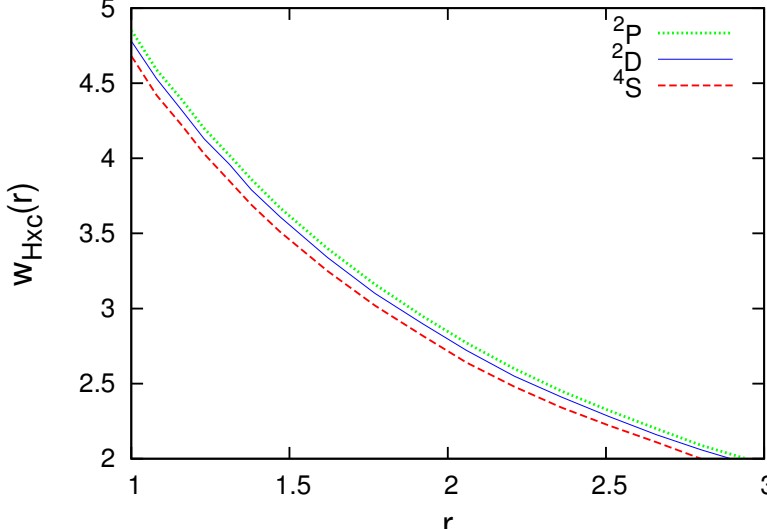

**Figure 2.** Subspace Hartree plus exchange-correlation potential for $^4S$, $^2D$ and $^2P$ of the N atom in atomic units (colored lines). The part of Figure 1, where the differences are the biggest, is enlarged.

Figures 3–7 present the subspace exchange-correlation potentials $w_{xc}$ for the B, C, N, O and F atoms. $w_{xc}$ has the same shape in all cases; $w_{xc}$ has a potential bump (or cusp). The OEP and KLI exchange potentials display the bump, in contrast to the local density approximations (LDA). The subspace KLI exchange potentials for $^3P$, $^1D$ and $^1S$ of the C atom have been presented in Ref. [5]. These curves also showed a potential bump. The present KLI-like plus local Wigner approach preserves this non-local character as the exchange dominates the correlation. All subspace exchange-correlation potentials presented exhibit the correct asymptotic behavior.

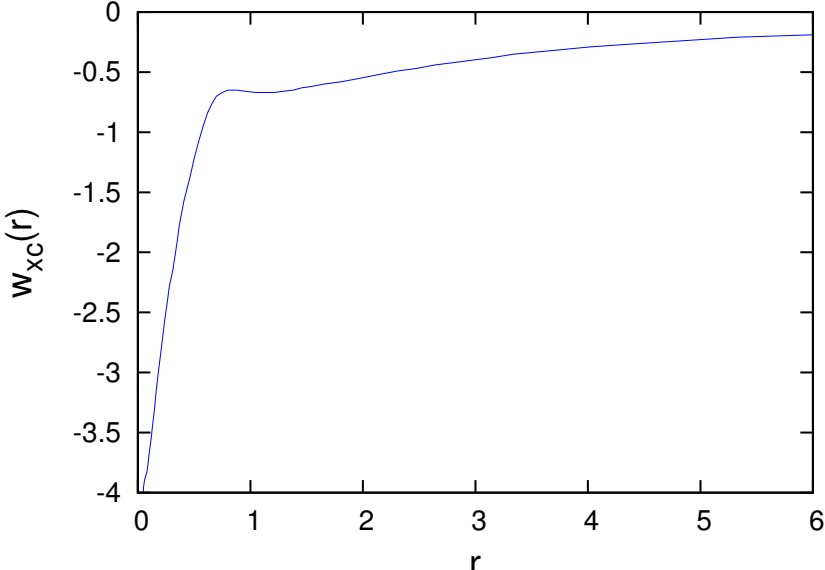

**Figure 3.** Subspace exchange-correlation potential of the B atom in atomic units.

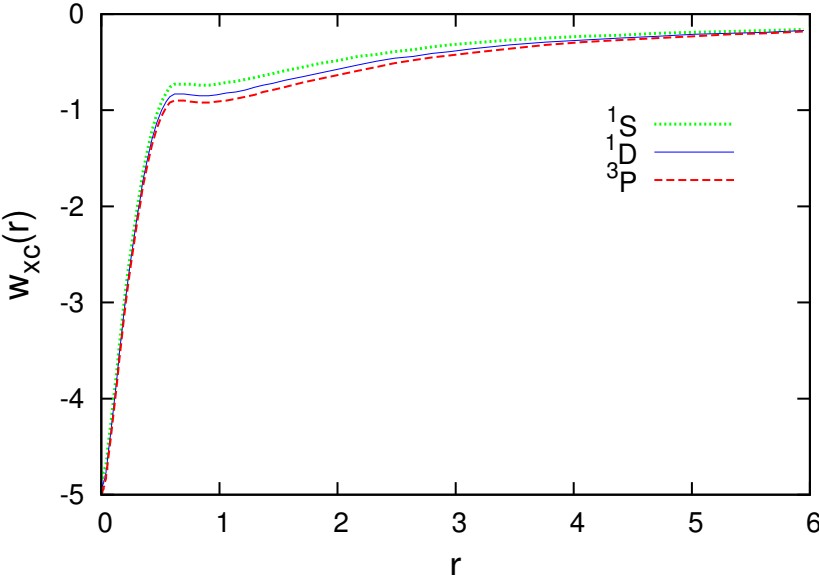

**Figure 4.** Subspace exchange-correlation potential for $^3P$, $^1D$ and $^1S$ of the C atom in atomic units.

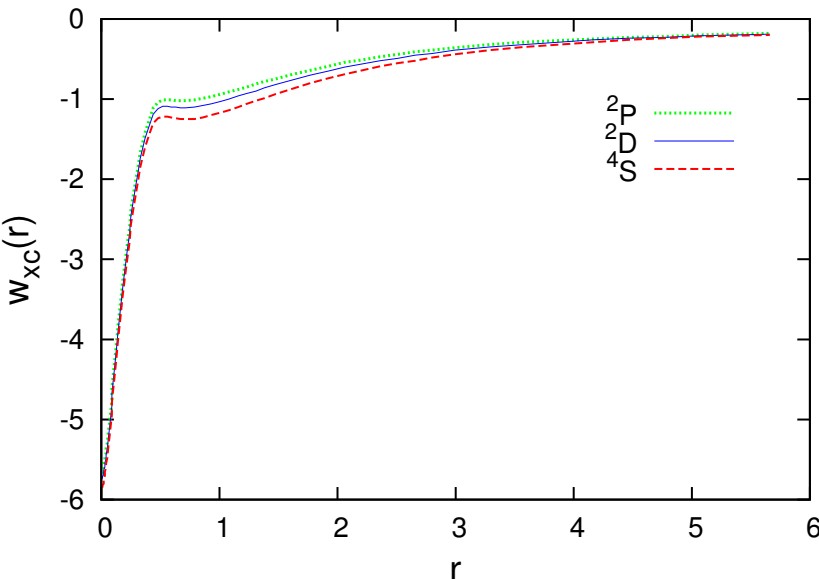

**Figure 5.** Subspace exchange-correlation potential for $^4S$, $^2D$ and $^2P$ of the N atom in atomic units.

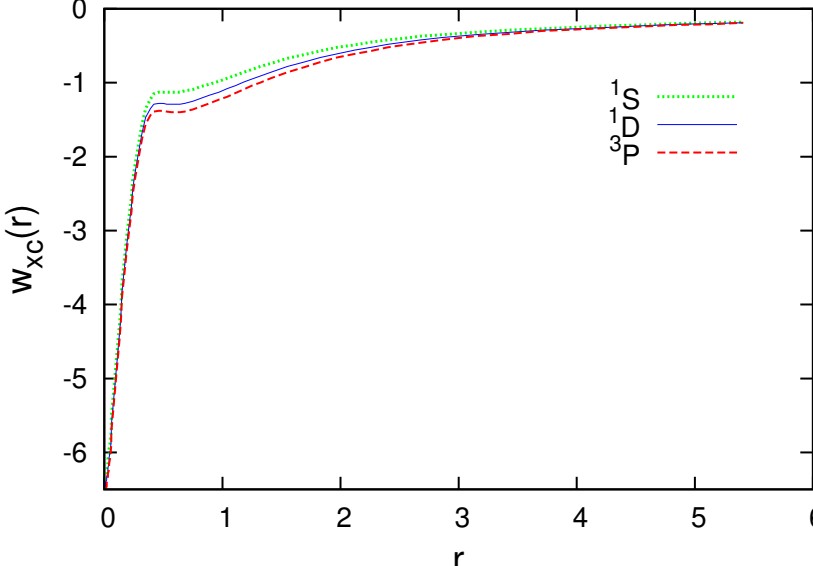

**Figure 6.** Subspace exchange-correlation potential for $^3P$, $^1D$ and $^1S$ of the O atom in atomic units.

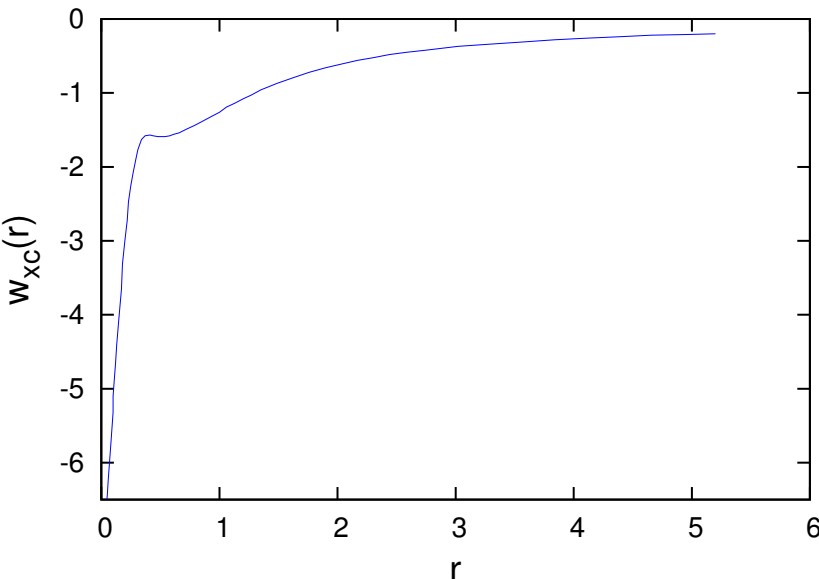

**Figure 7.** Subspace exchange-correlation potential of the F atom in atomic units.

## 4. Discussion

It should be emphasized that the functional $\mathcal{E}_{\{v\}}[\{w\}]$ is different from the functional defined by Yang, Ayers and Wu [7], because the variables are different; that is, their functional depends on the potential, while $\mathcal{E}_{\{v\}}[\{w\}]$ is a functional of the set $\{w\}$. However, of course, both functionals take the same value at the true ground-state. The same assertion is valid for the functional $\tilde{\mathcal{E}}_{\{v\}}[\{w_s\}]$. Therefore, the total energy $E$ and other important global quantities, such as the electronegativity or the hardness should also be the same as in the traditional DFT. This assertion is true for the local quantities such as local softness, as the present theory should yield the same density as the original DFT.

It is well-known that the density is almost spherically symmetric close to a nucleus and shares similarities with an atomic density. On the other hand, the density is spherically symmetric very far from the nuclei. It has been shown that each member of the set obeys a spherically symmetric Schrödinger-like Equation [31] which is equivalent to the Euler equation of this spherically symmetric density. The effective potential of this equation has been expressed in terms of wave function expectation values.

The asymptotic behavior of the density has been studied by several authors [32–35]. In Ref. [35], a differential Schrödinger inequality was derived and applied to determine the decay of the density and the spherically averaged density. The asymptotic decay of any member of the set of the spherically symmetric densities is given by [32–35]

$$\lim_{r_\beta \to \infty} \frac{\partial \bar{\varrho}_\beta(r_\beta)}{\partial r_\beta} = -\sqrt{8(E_0^{N-1} - E_0^N)}, \tag{50}$$

where the difference $E_0^{N-1} - E_0^N$ is the vertical ionization energy. $E_0^N$ and $E_0^{N-1}$ are the ground-state energies of the $N$- and $N - 1$-electron systems. That is, every member of the set decays in the same way.

The set of spherical densities bears some resemblance to the concept of 'atoms in molecules' (AIM) [31]. According to the most well-known AIM concept of Bader and coworkers [36], the molecule is divided into non-overlapping regions with one nucleus inside in each of them. The boundary of an atomic region is chosen so that the normal component of the density gradient is zero. In contrast to the free atom, 'an atom in a molecule' is 'closed' in an atomic region. The density averaged spherically around a nucleus, on the other hand, is not limited to an atomic region, it continues to infinity,

decaying as the ionization energy governs (Equation (50)). The integral of a spherically symmetric density

$$N = 4\pi \int \bar{\varrho}_\beta(r_\beta) r_\beta^2 dr_\beta \tag{51}$$

is the total number of the electrons of the molecule for all $\beta$, that is, for any member of the set. Obviously, it deviates from the number of electrons in an atomic region. Nevertheless, close to a nucleus the density is roughly spherically symmetric, therefore each spherically symmetric density shares similarities with an atomic density. Hence, a member of the set possesses some features of an atom, while other properties bear resemblance to a molecule. Consequently, though the present approach shows some similarity to AIM, it manifests a rather different concept.

Though traditional DFT is an exact theory, the exchange-correlation functional is not exactly known. Therefore, we have to employ approximations in computations. This assertion is true for the spherical theory, too. In SSPFT we have functionals of the set of the spherical potentials. Based on the success of the OEP-like methods, we can expect that the approximations making use of the spherical potentials will be valuable. Here, a KLI-like approximation is proposed as an illustration. Hopefully, using a more accurate correlation functional would improve this approach.

The spherical potentials have been relevant in the band structure calculations for a long time [37–39]. The muffin-tin approximation proposed by Slater [37] made use of the fact that the density is almost spherically symmetric in the vicinity of the nuclei. The Exact Muffin-Tin Orbitals (EMTO) Method is a powerful tool of calculations in solid-state physics and materials science (see details in the book by Vitos [40]).

Employing the spherically symmetric subspace potentials can yield even more efficient and powerful approaches. Therefore, SSPFT is expected to be powerful for computation. An effective potential having the form of (18) has already been proposed by Theophilou and Glushkov [41]. They introduced it as a direct mapping of the external potential. Their expression is

$$V_{TG} = -\sum_{\beta=1}^{M} \left[ \frac{Z_\beta}{r_\beta} + C_\beta \frac{1 - e^{\alpha_\beta r_\beta}}{r_\beta} \right], \tag{52}$$

where the parameters $C_\beta$ and $\alpha_\beta$ were determined by minimizing the Hartree–Fock energy. This intuitive expression gave reasonably good results for atoms and molecules [42,43].

The spherical density functional theory can be combined with a previous method to construct an orbital-free density functional theory [44]. It is possible to establish auxiliary spherical non-interacting (Kohn–Sham-like) systems. The set of the spherically symmetric densities can be extended to the generating spherical functions having two extra variables besides the radial distance from the centers. These generating functions can be used to calculate the Pauli potentials and then solve the Euler equations.

In summary, the recently initiated form of DFT using a set of the spherically symmetric densities has a 'set-representability problem'. We cannot be sure that there exists a density for a given set of the spherically symmetric densities. This 'set-representability problem' disappears if potentials are applied instead of densities as basic variables. In the case of degeneracy, spherically symmetric subspace potentials are proposed. These potentials are favorable because the subspace densities have the symmetry of the external potential if equal weighting factors are applied in the construction of the subspace. In atom, for example, the subspace density is exactly spherically symmetric. This version of the theory might offer more effective computation.

**Funding:** This research was supported by the National Research, Development and Innovation Fund of Hungary, financed under 123988 funding scheme.

**Institutional Review Board Statement:** Not applicable.

**Informed Consent Statement:** Not applicable.

**Data Availability Statement:** Data sharing is not applicable to this article.

**Conflicts of Interest:** The author declares no conflict of interest.

## Appendix A

For the readers' convenience, the proofs of the two main theorems of the subspace theory are presented in case of Coulomb external potential. First, the proof of Theophilou's theorem based on Kato's theorem is summarized.

**Theorem A1** (Theophilou's theorem). *The set of the spherically symmetric subspace densities $\bar{\varrho}_\beta(r_\beta)$ ($\beta = 1, \ldots, M$) determines uniquely the external potential if the external potential has the form of Equation (7).*

**Proof of Theophilou's theorem utilizing Kato's theorem.** Kato's theorem for the eigendensities of the Hamiltonian (7) [45–47] has the following form:

$$\left.\frac{\partial \bar{\varrho}_\beta^\gamma(r_\beta)}{\partial r_\beta}\right|_{r_\beta=0} = -2Z_\beta \varrho_\beta^\gamma(\mathbf{r} = \mathbf{R}_\beta) . \tag{A1}$$

It has been shown that [48] $\varrho_\beta^\gamma(\mathbf{r} = \mathbf{R}_\beta) = \bar{\varrho}_\beta^\gamma(r_\beta = 0)$. Therefore,

$$\left.\frac{\partial \bar{\varrho}_\beta^\gamma(r_\beta)}{\partial r_\beta}\right|_{r_\beta=0} = -2Z_\beta \varrho_\beta^\gamma(r_\beta = 0) . \tag{A2}$$

Using Equation (2), we can obtain Kato's theorem for the subspace density

$$Z_\beta = -\frac{1}{2} \frac{1}{\bar{\varrho}_\beta(r_\beta = 0)} \left.\frac{\partial \bar{\varrho}_\beta(r_\beta)}{\partial r_\beta}\right|_{r_\beta=0} . \tag{A3}$$

If the set of the spherically symmetric subspace densities is known, Equation (A3) yields the atomic numbers. The cusps of the members of the set provide the positions of the nuclei. The integral of any spherically symmetric subspace density offers the number of electrons. Therefore, all parameters of the external potential are known. That is, the set of the spherically symmetric subspace densities $\bar{\varrho}_\beta(r_\beta)$ ($\beta = 1, \ldots, M$) uniquely determines the external potential if the external potential has the form of Equation (7). □

In the spherical density functional theory (SDFT), there exists a 'set-representability problem'; that is, it is not certain that we can always find a density for a given set of the spherically symmetric densities. In the spherical potential functional theory (PDFT), on the other hand, there is no 'set-representability problem'.

**Theorem A2** (Set theorem in SSPFT). *There exists a one-to-one map between the ground-state potential $w$ and the set of the spherically symmetric potentials $\{w\}$ if it is known that $w$ has the form of Equation (7):*

$$w(\mathbf{r}) = \sum_{\beta=1}^{M} w_\beta(r_\beta), \tag{A4}$$

*where*

$$w_\beta(r_\beta) = -\frac{Z_\beta}{r_\beta} . \tag{A5}$$

**Proof of the set theorem in SSPFT in the case of Coulomb potential.** The proof of the set theorem is very simple in the case of Coulomb potential. The proof that the set $\{w\}$ determines the potential $w$ is trivial, as Equation (A4) yields $w$ if the set $\{w\}$ is known. The proof that the potential $w$ determines the set $\{w\}$ is also simple if we have Coulomb potential. (The proof for the general case of the form of Equation (7) is more complicated and can be found in Ref. [8].) We know that $w$ has the form of Equations (A4) and (A5), therefore $w$ determines the positions of the nuclei; these are in the points $\mathbf{R}_\beta$, where $w$ tends to minus infinity. The atomic numbers are given by the limits

$$Z_\beta = -\lim_{\mathbf{r}\to\mathbf{R}_\beta}(|\mathbf{r}-\mathbf{R}_\beta|)w(\mathbf{r}-\mathbf{R}_\beta). \tag{A6}$$

That is, Equation (A5) provides the members of the set $\{w\}$. $\quad\square$

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
