# Peer review of "Spherical Subspace Potential Functional Theory"

_computation, doi:10.3390/computation11060119_

Round 1

Reviewer 1 Report

In this article, the author summarizes previous developments on the theory of subspace with spherically symmetric densities, where atoms can be treated as spheres when using a subspace density. The representability problem is solved by using potential instead of density. This potential functional theory utilizes the special form of the external potential. An application to the Nitrogen atom is shown. The article is very well written and can be followed despite its difficulty. In my opinion it deserves to be published, but I have two suggestions for the author:

11)      Perform more atom calculations. N atom (in this paper) and C atom (in JCP 154, 074103,2021) have been presented. It will be interesting to compare the exchange-correlation potentials for the p series of atoms and discuss them.

22)      Add some comment in the discussion about the difference with Atoms in Molecules  theory (AIM), may also be stimulating.

Correction: Labels of Figs 1 and 2, 1D should be 2D.

Author Response

I am grateful to the referee for the valuable comments and accordingly I have made changes in the revised version

11) I added results, figures and discussion of  the exchange-correlation 
potentials for the p series of atoms from B to F to Section III. 

22)  Comment in the discussion about the difference with Atoms
 in Molecules  theory (AIM) is added to Section IV. 

Labels are corrected.

Reviewer 2 Report

The subject of the article is an interesting approach for those who develop DFT methods and computational chemistry. Remarks: - Figure 1 is not very clear, the representations overlap a lot, I would recommend using different symbols;; - in the graphic representations, is the distance r expressed in atomic units or Angstroms?  

The quality of the English Language is good

Author Response

I am grateful to the referee for the valuable comments and accordingly I have made changes in the revised version.

 A part of Figure 1 is enlarged and added as Figure 2 in order to improve the visibility of the curves. All figures are in atomic units. It is added to captions.

Reviewer 3 Report

The work is a sound development in basic issues regarding DFT in atoms.

One may publish as it is. However, as optional suggestion, I think it will be useful to elaborate a bit more around  eq. 47: few words about the principles of its proof (although citations are given) and maybe a discussion about relation with electronegativity and hardness.

Author Response

I am grateful to the referee for the valuable comments and accordingly I have made changes in the revised version.

Several sentences are added to the Discussion around  eq. 47. including
the starting point of the proof. A remark on electronegativity and hardness
is also added.